THE NATURAL HISTORY OF MODEL ORGANISMS

# From molecular manipulation of domesticated *Chlamydomonas reinhardtii* to survival in nature

**Abstract** In the mid-20th century, the unicellular and genetically tractable green alga *Chlamydomonas reinhardtii* was first developed as a model organism to elucidate fundamental cellular processes such as photosynthesis, light perception and the structure, function and biogenesis of cilia. Various studies of *C. reinhardtii* have profoundly advanced plant and cell biology, and have also impacted algal biotechnology and our understanding of human disease. However, the 'real' life of *C. reinhardtii* in the natural environment has largely been neglected. To extend our understanding of the biology of *C. reinhardtii*, it will be rewarding to explore its behavior in its natural habitats, learning more about its abundance and life cycle, its genetic and physiological diversity, and its biotic and abiotic interactions.
DOI: https://doi.org/10.7554/eLife.39233.001

**SEVERIN SASSO\*, HERWIG STIBOR, MARIA MITTAG AND ARTHUR R GROSSMAN\***

\*For correspondence: severin.
sasso@uni-jena.de (SS); arthurg@
stanford.edu (ARG)

**Competing interests:** The authors declare that no competing interests exist.

## Introduction

*Chlamydomonas reinhardtii* is a single-celled green alga found in temperate soil habitats (*Figure 1*). It has proven to be such a powerful model for dissecting fundamental processes in biology that investigators have dubbed it the 'green yeast' (*Goodenough, 1992*; *Rochaix, 1995*). Ehrenberg described the genus *Chlamydomonas* in 1833, and Dangeard the species *C. reinhardtii* in 1888 (*Harris et al., 2009*). *Chlamydomonas* was found suitable for genetic studies in the early 20th century (*Harris, 2001*), while the development of *C. reinhardtii* as a model organism dates to the 1950s when the first mutants were generated (*Harris, 2009*).

Various features make *C. reinhardtii* an excellent laboratory species. It grows vegetatively as a haploid, which allows mutant phenotypes to be expressed immediately. Under optimal conditions, *C. reinhardtii* grows so rapidly that its numbers can double approximately every 8 hours (*Harris, 2001*). The fact that it can grow in the dark on acetate-containing medium while retaining a functional photosynthetic apparatus, has allowed even light-sensitive photosynthesis

mutants to be isolated (*Levine, 1969*; *Spreitzer and Mets, 1980*). The motile cilia of this photosynthetic eukaryote share the same structure and many of the same constituent proteins as those of mammals, and so research into its motility prompted studies that greatly advanced our understanding of cilium dysfunctions in humans (*Brown and Witman, 2014*). Furthermore, *C. reinhardtii* can be induced to sexually reproduce in the laboratory, making it easy to introduce multiple traits into a single haploid strain (e.g. to generate double or triple mutants). The power of *C. reinhardtii* as a model organism was further elevated by the advent of genetic transformation (*Boynton et al., 1988*; *Kindle, 1990*; *Remacle et al., 2006*), the establishment of a full nuclear genome sequence (*Merchant et al., 2007*), the construction of a genome-wide library of mapped, indexed insertional mutants (*Li et al., 2016*) and CRISPR-mediated targeted gene disruptions (*Ferenczi et al., 2017* and references therein).

Studies of *C. reinhardtii* have enabled numerous landmark discoveries and advances. One remarkable example is the discovery of intraflagellar transport of granule-like particles

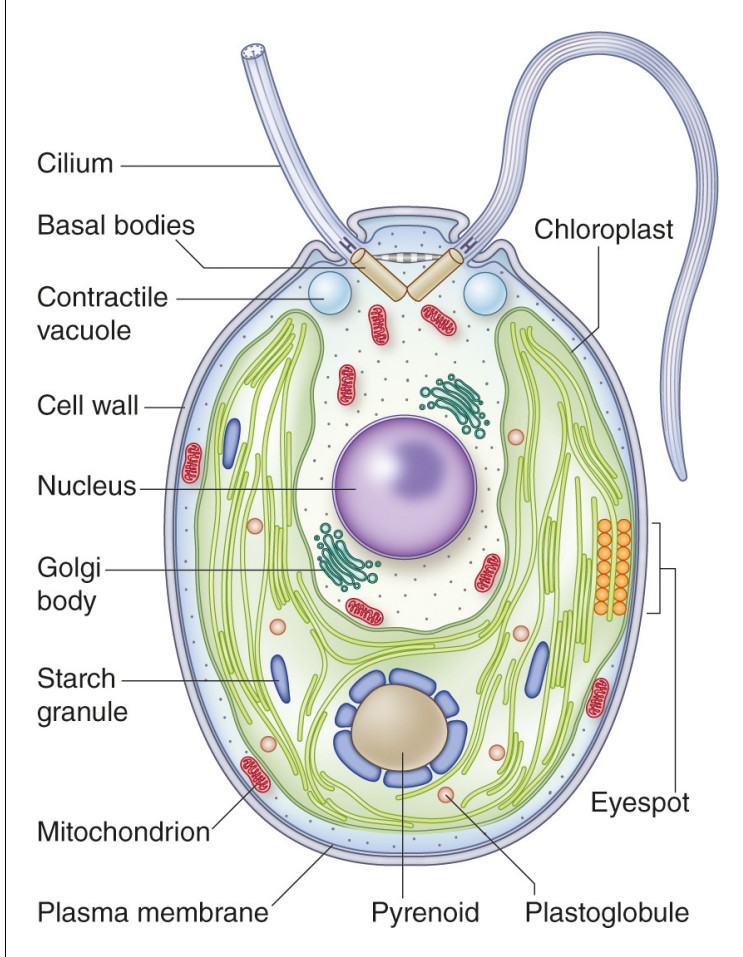

**Figure 1.** Structure of a vegetative *Chlamydomonas reinhardtii* cell. This cell has a 5-10 μm diameter (*Gallaher et al., 2015*). The two anterior cilia possess a 9+2 microtubule structure characteristic of motile cilia of eukaryotes. The cilia are critical for mating processes and confer motility to the cell (*Harris, 2001*). A single cup-shaped chloroplast occupies a large proportion of the cell's volume. This organelle houses the machinery for oxygenic photosynthesis and contains the pyrenoid, a structure in which Rubisco is concentrated; the pyrenoid is a component of the carbon concentrating mechanism (CCM) which functions to concentrate inorganic carbon in the cell against a concentration gradient (*Mackinder et al., 2016*). Close to the cell equator, at the edge of the chloroplast, is the eyespot. This primordial visual system allows the cells to orient their swimming toward or away from the light (phototaxis). Under hypoosmotic conditions, the cytoplasmic water content is maintained by pumping water out of the cell through contractile vacuoles positioned at the cell's anterior (*Komsic-Buchmann et al., 2014*). At the base of the cilia are the basal bodies, which are responsible for ciliary assembly (*Dutcher and O'Toole, 2016*). Other features of the cell include a centrally located nucleus, a proteinaceous cell wall, Golgi bodies within the cup-shaped region formed by the chloroplast, and mitochondria. Image credit: Debbie Maizels.
DOI: https://doi.org/10.7554/eLife.39233.002

(*Kozminski et al., 1993*) and the roles of motor proteins in the process (*Prevo et al., 2017* and references therein). Furthermore, structural analyses of wild type and mutants with defective cilia have massively contributed to our knowledge of the building blocks of these structures, their organization and their function (*Goodenough and Heuser, 1985*; *Silflow and Lefebvre, 2001*; *Nicastro et al., 2006*). These

analyses also led to classic studies that demonstrated that abnormal cilia could cause human genetic diseases such as polycystic kidney disease (*Pazour et al., 2000*; *Li et al., 2004*). Additionally, acetate-requiring mutants (often unable to perform photosynthesis) have immensely advanced our understanding of photosynthesis, especially the ordering of electron carriers in the photosynthetic electron transport chain (e.g.

*Gorman and Levine, 1965*). Two core proteins of photosystem II (D1 and D2) were first identified in *C. reinhardtii* (*Chua and Gillham, 1977*) and later proposed to be key components of this photosystem's reaction centers (*Deisenhofer et al., 1985*; *Trebst, 1986*; *Satoh, 2003*). More recently, a central role of the STT7 kinase in photosynthetic state transitions (*Depège et al., 2003*), and a key function of the xanthophyll cycle in nonphotochemical quenching were first established in *C. reinhardtii* (*Niyogi et al., 1997*).

The field of optogenetics experienced a recent quantum leap with the discovery of channelrhodopsins in *C. reinhardtii*. When expressed in other cells, these gated ion channels can be stimulated with light to activate various processes, including neuronal activity (*Hegemann and Nagel, 2013*). Sophisticated genetic, biochemical and cell biological analyses of *C. reinhardtii* are currently being performed to understand the cell cycle (*Cross and Umen, 2015*), basal bodies/centrioles function (*Dutcher and O'Toole, 2016*), pyrenoid structure (*Freeman Rosenzweig et al., 2017*),

mechanisms associated with photoreceptor function and light acclimation (*Minagawa and Tokutsu, 2015*; *Petroutsos, 2017*) and organismal interactions in ecosystems (*Thrane et al., 2016*). Finally, *C. reinhardtii* is being exploited to study the evolution of multicellularity, especially with respect to multicellular algal species of the order Volvocales (*Hallmann, 2011*).

## Habitats and biogeography

*C. reinhardtii* can unambiguously be identified by sequencing internal transcribed spacers (ITS) or various phylogenetically informative genes (*Pröschold et al., 2005*). Yet many ecological studies have relied on light microscopy to identify *Chlamydomonas* species (*sensu lato* – see *Box 1*). Typically, two anterior cilia and a cup-shaped chloroplast harboring a pyrenoid have been sufficient criteria for a cell to be considered a *Chlamydomonas* sp. This morphology-based identification may be reliable to the level of genus, but rarely to the species level since many species look very similar. For these reasons, at times we omit species designations and simply note the organism as *Chlamydomonas*

## Box 1. Taxonomic and laboratory history of *C. reinhardtii*

Based on traditional taxonomic criteria, the genus *Chlamydomonas* (*sensu lato*) contains more than 500 species. In the course of taxonomic revisions, which are still in progress, *Chlamydomonas* (*sensu stricto*) is comprised of three species (*Pröschold et al., 2018*). Accordingly, the taxonomy of some species mentioned in this article, such as *C. nivalis* or *C. euryale*, may be revised in the future. Furthermore, our use of the designations 'Chlamydomonas sp.' and 'Chlamydomonas spp.' refers to one or more *Chlamydomonas* species, respectively, which were typically not classified to the level of species and may not be *C. reinhardtii*. The majority of the contemporary *C. reinhardtii* laboratory strains were derived from a single zygote isolated from a potato field in Massachusetts in 1945 (*Harris, 2009*). The sequencing of 39 common laboratory strains shows that they fall into five genetically distinct lineages from two parents or haplotypes (*Gallaher et al., 2015*). Under laboratory conditions, mutations accumulate at a rate of ~0.03 division$^{-1}$ genome$^{-1}$, corresponding to one mutation every 30 generations (*Gallaher et al., 2015*). In addition, removal of *C. reinhardtii* from its natural environment, including cultivation in the laboratory or cryopreservation, may unintentionally select for specific traits. For example, *C. reinhardtii* is often grown on medium containing ammonium as a nitrogen source, which allowed for the evolution of mutants (*nit1*, *nit2*) unable to utilize nitrate (*Harris, 2009*; *Gallaher et al., 2015*). For these reasons, the isolates domesticated for decades in the laboratory may only loosely correspond to wild *C. reinhardtii* strains. Furthermore, we do not know if the laboratory strains are still capable of surviving in the wild. To examine the ecological significance of laboratory findings, it will be important to isolate additional wild *C. reinhardtii* strains and characterize their behavior both in the field and in culture.

DOI: https://doi.org/10.7554/eLife.39233.003

sp. A routine use of genetic taxonomic markers in the future would improve our knowledge of the geographical distribution of *C. reinhardtii* and related species and allow for more precise classifications.

While *Chlamydomonas* spp. (not identified at the species level) occur widely in temperate, subtropical and tropical soils (*Starks et al., 1981*), confirmed *C. reinhardtii* has only been found in temperate soils in Northern America and Japan (*Pröschold et al., 2005*; *Nakada et al., 2010*). It occurs in cultivated fields but appears absent from many other habitats, suggesting it prefers nutrient-rich, disturbed soils (*Sack et al., 1994*). Most contemporary laboratory strains have emanated from a single soil isolate collected in 1945 (*Box 1*). Light typically penetrates only millimeters into the soil, depending on factors such as the soil structure and moisture content

(*Tester and Morris, 1987*; *Ciani et al., 2005*). Therefore, photosynthetic microbes are generally most abundant in the upper few millimeters where they can harvest light energy, although in some instances they can be present in soil layers where there is essentially no light (*Metting, 1981*). *Chlamydomonas* spp. are even present in biological soil crusts where they help stabilize the surface of drylands, contribute to primary production and potentially act as pioneer species (*Büdel et al., 2009*).

All unambiguously identified *C. reinhardtii* isolates were collected from soil habitats (T. Pröschold, personal communication), yet *Chlamydomonas* spp. are also commonly found in the pelagic zone of lakes, where they sometimes form spring blooms (*Similä, 1988*; *Krivtsov et al., 2000*). The term 'pelagic zone' refers to the water column of lakes and oceans not on or near the lake or ocean bottom.

---

## Box 2. Outstanding questions about the natural history of *C. reinhardtii*

- What are the geographic origins of *C. reinhardtii*? What are its current geographic and vertical distributions? How do populations of *C. reinhardtii* quantitatively change over time and what factors impact these changes? What are the major mechanisms of *C. reinhardtii* dispersal? For example, are aquifers common routes for the transport of *C. reinhardtii* over long distances?

- What is the genetic variability within and between *C. reinhardtii* populations? What are the relationships among populations of the various *Chlamydomonas* species?

- Do specific pelagic strains of *C. reinhardtii* exist in lakes? If so, do they have major differences in their life histories, physiologies and genome sequences compared to soil-dwelling strains?

- What are the typical division rates of vegetative *C. reinhardtii* cells in the wild? How frequently does sexual reproduction occur in natural populations? How common are dormant zygospores in the environment, and where do they occur? Are zygospores typical overwintering forms, and do they also have an increased resistance to challenging biotic interactions?

- What are the most common biotic interactions of *C. reinhardtii* in the environment (competing photosynthetic microbes, grazers, bacteria, fungi)? In what ways does *C. reinhardtii* communicate with its neighbors (e.g. infochemical signals)? What is the metabolic significance of these interactions?

- How often and under what situations do cells shed their cilia in nature? Is there a selective advantage of deciliation in response to stress?

- Does *C. reinhardtii* associate with biofilms on soil particles and, if so, how are the algal cells organized within the biofilm community?

DOI: https://doi.org/10.7554/eLife.39233.004

---

*Chlamydomonas* spp. are usually motile, and although this has an energetic cost, it gives them a competitive advantage in lakes that have stratified into distinct layers as a consequence of seasonal changes in temperature (*Striebel et al., 2009*). Under conditions of stratification, motile algae often ascend toward the lake surface during the day to optimize their exposure to sunlight. During the night, they tend to descend to access the nutrient-rich environment below the surface. Indeed, this pattern of vertical movement has been observed for the population of *Chlamydomonas* sp. in a small Finnish lake (*Jones, 1988*).

Environmental conditions and the availability and distribution of natural resources differ substantially in soils and lakes (*Sommer et al., 2012*; *Coleman et al., 2017*). Phosphorous, for example, is likely to be limiting to the growth of organisms in lakes and geologically old soils, while nitrogen limitation is more common in young soils (*Schindler, 1977*; *Vitousek and Howarth, 1991*). Light availability and grazing pressure by predators represent additional key environmental differences between soil and lake habitats. Consequently, these two habitats require distinct adaptations and life history strategies to optimize fitness, and it is still an open question as to whether specific pelagic strains of *C. reinhardtii* exist in lakes (*Box 2*).

*Chlamydomonas* spp. other than *C. reinhardtii* are adapted to a wide range of habitats. For example, *Chlamydomonas eustigma* is an acidophilic species isolated from acid mine drainage (*Hirooka et al., 2017*), *Chlamydomonas euryale* is found in temperate marine environments (*Burch et al., 2015*), *Chlamydomonas* spp. have been isolated from Antarctic ice (*Liu et al., 2006*), and some members of the genus *Chlamydomonas* are carotenoid-rich organisms present on the surface of snow, giving it a red appearance (*Remias et al., 2005*). A *Chlamydomonas* sp. has even been identified in the air at 1,100 meters above the ground: this and other algae can be dispersed by wind over extended distances (*Brown et al., 1964*). Taken together, several reports provide information on the biogeographical distribution of *C. reinhardtii* and other *Chlamydomonas* spp. However, there is little knowledge of the abundance and variations of *Chlamydomonas* spp. in different soil types, the dynamics of these natural populations over daily or seasonal cycles, and their physiological capabilities.

## Genomics

The chloroplast and mitochondrial genomes of *C. reinhardtii* have been sequenced and are 206 and 15.8 kb, respectively (*Vahrenholz et al., 1993*; *Maul et al., 2002*; *Gallaher et al., 2018*). Since the sequence of the nuclear genome was first published (*Merchant et al., 2007*), the scientific community has focused some effort on elevating the quality of the genome sequence and improving its assembly and annotation (*Blaby et al., 2014*). The current version 5.5 nuclear genome is 111 Mb, which is similar in size to the genome of the model land plant *Arabidopsis thaliana* (*Blaby et al., 2014*). Recently, whole-genome sequences for more than 50 additional laboratory strains and field isolates were generated (*Flowers et al., 2015*; *Gallaher et al., 2015*). The sequences of 12 field isolates confirmed earlier reports that with a nucleotide diversity ($\pi$) of ~3%, the *C. reinhardtii* genome is among the most polymorphic of all eukaryotes (*Flowers et al., 2015*). The field strains, isolated from various locations in the United States and Canada, genetically group into three distinct populations, with gene flow between populations sufficiently low to allow the populations to adapt to their local environments. The low ratio of genome-wide non-synonymous to synonymous substitutions (0.58) further indicates that natural selection efficiently eliminates *C. reinhardtii* alleles of low fitness (*Flowers et al., 2015*). Whole-genome and epigenome sequencing has also been used to examine adaptation in the laboratory under changing environmental conditions (*Kronholm et al., 2017*).

## Life cycle and its role in nature

Forming zygotes likely allows *C. reinhardtii* to survive when conditions become harsh (*Harris, 2001*; *Goodenough et al., 2007*). In the laboratory, gametogenesis can be induced by nitrogen starvation (*Treier et al., 1989*) in conjunction with specific light conditions; both signals may inform the cell of deteriorating environmental conditions (see below). The fusion of haploid gametes results in diploid zygotes that can develop over several days into highly resistant, dormant zygospores (*Figure 2*). When nitrogen is added back to the medium, the zygotes germinate in the light, undergo meiosis and typically release four haploid cells that resume vegetative growth (*Harris, 2001*). Dormant zygospores can remain viable in soil for many years (*Harris, 2001*) and survive freezing (*Suzuki and Johnson, 2002*), desiccation (*Heimerl et al.,*

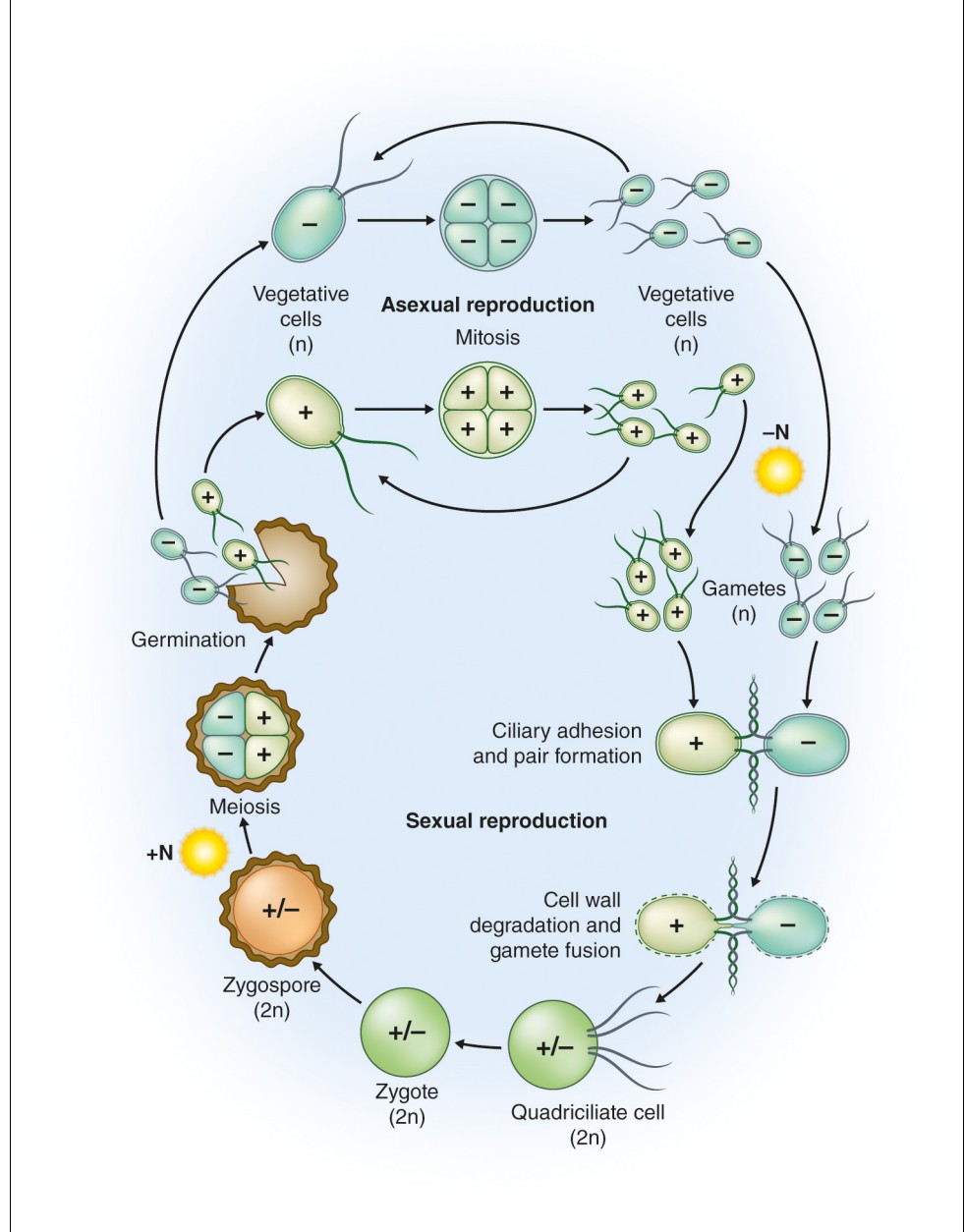

**Figure 2.** Life cycle of *C.reinhardtii*. Haploid (n) vegetative cells occur as two mating types, $mt^+$ and $mt^-$, that divide by mitosis ("Asexual reproduction"; ***Harris, 2001***, ***Goodenough et al., 2007***). Gametogenesis can be induced by nitrogen starvation (-N) in the presence of light, and gametes of opposite mating types can fuse to form diploid (2n) zygotes ("Sexual reproduction"). Within a few hours of fertilization, zygotes resorb their four cilia to become immotile. Over the course of several days these zygotes are remodeled into highly resistant, dormant zygospores. In this process, a strong, multilayered cell wall is formed, and chlorophyll is degraded (***Harris, 2001***; ***Goodenough et al., 2007***). As a result, mature zygospores appear orange, which reflects their carotenoid content (***Lohr, 2009***). When environmental conditions improve, the zygote undergoes meiosis to release four haploid cells (sometimes 8 and 16 when mitosis also occurs within the zygote wall; "Germination"). The haploid cells then resume vegetative growth. In the laboratory, zygote germination is induced by the addition of nitrogen (+N) to the medium in the light (***Harris, 2001***); nitrogen also causes reprogramming of gametes to vegetative cells (***Pozuelo et al., 2000***). Image credit: Debbie Maizels.

DOI: https://doi.org/10.7554/eLife.39233.005

2018) and probably other forms of harsh environmental conditions. This extraordinary resistance is associated with the multilayered cell wall of the zygospores, which contains a durable lipid polymer structurally similar to those found in million-year-old microfossils (described for *Chlamydomonas monoica*; *Blokker et al., 1999*). Furthermore, sexual reproduction can increase the rate of adaptation of *C. reinhardtii* to new or changing environmental conditions, particularly if the population and genetic diversity within the population are large (*Colegrave, 2002*).

Following gamete fusion, a pair of homeodomain transcription factors initiates the genetic program for zygote development (*Kurvari et al., 1998*; *Lee et al., 2008*). The first zygote-specific genes are induced within minutes of gamete fusion, with hundreds of additional genes activated over the next few hours (*Lopez et al., 2015*; *Joo et al., 2017*). A gene encoding a polyketide synthase is induced two days after zygote formation and is critical for the zygote-to-zygospore transition, probably because it participates in the biosynthesis of the cell wall lipid polymer (*Heimerl et al., 2018*). Several stages of the sexual cycle, including gamete formation and maintenance and zygote germination, depend on light and involve regulation by three different photoreceptors (*Huang and Beck, 2003*; *Müller et al., 2017*; *Zou et al., 2017*).

Similar to *Saccharomyces cerevisiae* (*Liti, 2015*), we know little about the life cycle of *C. reinhardtii* in its natural environment. For example, there is no quantitative data on the frequency of sexual reproduction relative to vegetative growth. Yet, nitrogen is thought to become limited more commonly in soils than lakes (*Schindler, 1977*; *Vitousek and Howarth, 1991*; *Coleman et al., 2017*). This notion is congruent with nitrogen limitation being a major cue for zygospore formation in nature, but we are not aware of any data on zygospore induction in natural soil environments. The occurrence of clonal cultures of opposite mating types that are derived from a single zygospore isolated from dry soil provides additional evidence for a critical role of zygospores during desiccation (*Harris, 2009*). Freezing resistance of zygospores and their more efficient germination under long-day conditions compared to short-day conditions suggests that zygospore formation is an overwintering strategy (*Suzuki and Johnson, 2002*). If true, the question arises as to whether or not nitrogen limitation and day length are adequate cues to herald the approach of winter, or whether, for example, a decrease in soil moisture content or temperature can also induce zygospore formation in *C. reinhardtii*.

## Physiological and metabolic capabilities

*C. reinhardtii* not only orients itself with respect to light, but can also swim upward in complete darkness. This negative gravitaxis may facilitate orientation and movement of the cells at night or in the soil environment, potentially helping the alga locate areas with more favorable conditions of illumination following daybreak (*Bean, 1977*). Furthermore, vegetative cells are attracted to ammonium, nitrite and nitrate (*Ermilova and Zalutskaya, 2014* and references therein). Chemotaxis towards ammonium is strongest during the night, whereas phototaxis towards the light is strongest during the day, with both processes regulated by the circadian clock (*Bruce, 1970*; *Byrne et al., 1992*). Finally, the hypothesis that the circadian clock depends on gravity or a magnetic field was refuted by experiments performed with *C. reinhardtii* on a space shuttle under microgravity conditions (*Mergenhagen and Mergenhagen, 1987*).

Cilia enable *C. reinhardtii* to swim in an aqueous medium, and also glide on solid surfaces. Gliding motility may be important when *C. reinhardtii* resides within a thin water film that coats soil particles (*Mitchell, 2000*). The gliding speed of *C. reinhardtii* is ~1 $\mu$m s$^{-1}$ (*Shih et al., 2013*) whereas the average forward swimming speed is 100-200 $\mu$m s$^{-1}$ (*Rüffer and Nultsch, 1985*). Under various stress conditions, such as acidification of the medium, *C. reinhardtii* loses or sheds its cilia when a specific break point near the base of the cilium is activated (*Quarmby, 2009*). When conditions improve, the cilia regenerate. The biological reason for deciliation is still a mystery, but various hypotheses have been put forth (*Quarmby, 2009*). Deciliation is observed in a wide range of different cell types; for example, inhalation of irritant chemicals can lead to deciliation of respiratory epithelial cells in mammals (*Buckley et al., 1984*). Therefore, it seems likely that a predetermined break point is an ancient and inherent property of every cilium (*Quarmby, 2009*). Consequently, deciliation may not confer a selective advantage, but might be a consequence of pathological conditions that cause over-stimulation of the ciliary disassembly process. On the other hand, the ciliary membrane of *C. reinhardtii* is in direct contact with the environment (not

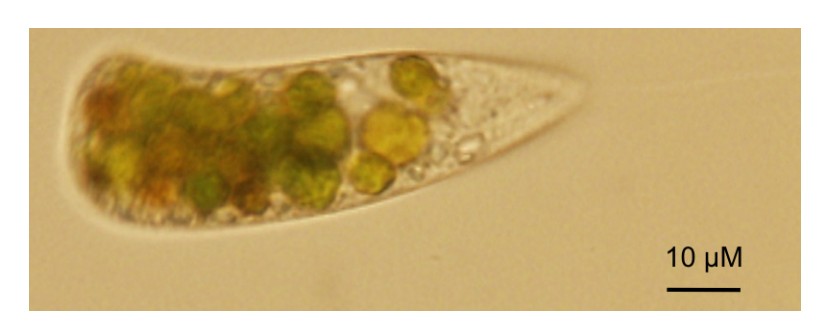

10 µM

**Figure 3.** *C. reinhardtii* ingested by the predatory protist *Peranema trichophorum*. Image credit: Santosh Sathe and Pierre Durand.
DOI: https://doi.org/10.7554/eLife.39233.006

protected by cell wall) and therefore, deciliation may reduce the entrance of noxious compounds into cells. Deciliation may also allow cells to escape when their cilia are stuck to the surface of a predator (*Quarmby, 2009*). Studying deciliation in the natural environment holds the promise of new insights into selection pressures that led to its evolution.

The ability of *C. reinhardtii* to grow under heterotrophic and fermentative conditions might be an adaptation to soil environments where there can be both low light and low oxygen. Anoxic/hypoxic conditions are likely to mostly occur at night when there is no photosynthesis to release oxygen and the soil microbes are respiring. Under anoxic conditions, *C. reinhardtii* can use glycolysis to yield energy, which is sustained by fermentation metabolism and the release of reduced organic compounds (*Catalanotti et al., 2013*). *C. reinhardtii* has recently been shown to activate a variety of different pathways that result in the formation of many fermentation products including formate, lactate, acetate, acetyl-CoA, succinate, hydrogen and glycerol (*Atteia et al., 2013*; *Catalanotti et al., 2013*; *Yang et al., 2015*). While some regulatory elements involved in anoxic metabolism are known (*Hemschemeier et al., 2013*; *Huwald et al., 2015*; *Düner et al., 2018*), little is understood about what controls the various pathways associated with fermentation and the ways in which these pathways are integrated.

## Biotic interactions

In nature, *C. reinhardtii* is continuously in contact with other organisms, including competitors, predators, pathogens, parasites, commensals or mutualists. Most molecular details concerning these interactions, which likely involve chemical

signaling, nutrient exchange and receptor-mediated processes, have not been examined. In lakes, the various *Chlamydomonas* spp. successfully compete with many other pelagic algal species for light and nutrients; rapid growth of *Chlamydomonas* spp. likely compensates for severe grazing losses, such as during periods of rapid proliferation of filter feeders, like water fleas (cladocerans). High rates of algal growth demand high nutrient levels. The concentrations of dissolved nutrients during the growing season are usually highest after spring mixing (*Sommer et al., 2012*), and therefore, the abundance of *Chlamydomonas* spp. in temperate lakes often shows a strong peak in spring or summer (*Dembowska, 2015*). In addition, the absence of filter feeders and the presence of more selective feeders in the soil may result in lower grazing losses and less seasonal differences in abundance patterns.

Predation of *C. reinhardtii* by zooplankton such as *Daphnia*, a highly efficient filter feeder (*Van Donk et al., 1997*), rotifers (*Lurling and Beekman, 2006*), and protists such as *Tetrahymena* (*Taub and McKenzie, 1973*) or *Peranema* (*Figure 3*), has been shown to occur in the laboratory. These predators either live exclusively in the pelagic zone of lakes, or at least more commonly in this habitat compared to soils. In the soil, animals such as earthworms or springtails, and protists are typical predators of microscopic algae (*Schmidt et al., 2016*; *Seppey et al., 2017*), but there is currently little specific information on predators of *C. reinhardtii*. The formation of large aggregates of *C. reinhardtii* cells is a general and probably non-specific defense strategy by which the alga may avoid ingestion. For example, the rotifer *Brachionus calyciflorus* triggers the formation of so-called palmelloid colonies (*Lurling and Beekman, 2006*). These colonies are aggregates of *C. reinhardtii* that may form as a consequence of the failure of the mitotically dividing mother cell to release the daughter cells from its encapsulating cell wall (*Khona et al., 2016*). This phenomenon may be triggered by stress under conditions in which zygospore formation is not possible (*Khona et al., 2016*). On the other hand, *C. reinhardtii* can actively aggregate in the presence of the predatory protist *Peranema trichophorum* (*Sathe and Durand, 2016*). A *P. trichophorum* culture filtrate was able to induce algal aggregation, suggesting that *C. reinhardtii* senses an unidentified substance (a kairomone) that is released by the predator (*Sathe and Durand, 2016*).

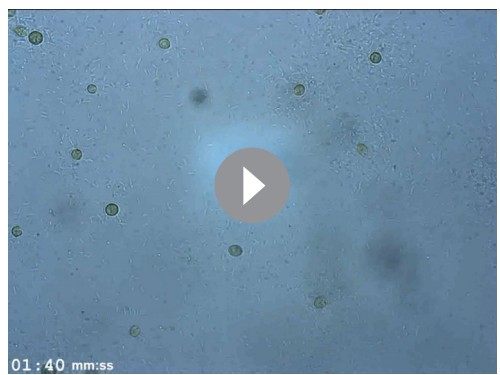

**Video 1.** *C. reinhardtii* surrounded by the harmful bacteria *Pseudomonas protegens* (*Aiyar et al., 2017*)
Video credit: Prasad Aiyar, Severin Sasso and Maria Mittag.
DOI: https://doi.org/10.7554/eLife.39233.007

C. reinhardtii is also a prey for soil bacteria. The bacterium *Pseudomonas protegens* can surround and immobilize algal cells (*Video 1*); it secretes a cyclic lipopeptide that triggers an increase in calcium levels inside *C. reinhardtii* cells with subsequent deciliation (*Aiyar et al., 2017*). This antagonistic interaction inhibits algal growth and probably leads to the death of most of the algal cells; the bacteria may acquire trace metals from the dying cells (*Aiyar et al., 2017*). Furthermore, small molecules from *C. reinhardtii* activate quorum sensing in *Pseudomonas aeruginosa* (*Rajamani et al., 2008*). It will be important to determine if algal cells also produce quorum-sensing mimics that influence *P. protegens*. Finally, while no viral pathogens of *C. reinhardtii* have been reported, it seems likely that they exist. The areas of algal-bacterial and algal-viral interactions are fertile for more probing basic research.

Several beneficial interactions of *C. reinhardtii* have been described, including interactions with growth-promoting bacteria and even mutualism (e. g. *Nikolaev et al., 2008*; *Lörincz et al., 2010*; *Kim et al., 2014*). These findings provide the basis for future studies that address regulatory mechanisms and identify specific compounds that impact the biology of *C. reinhardtii* in nature. One compound synthesized by prokaryotes and used by many algae is vitamin $B_{12}$. Although *C. reinhardtii* does not depend on vitamin $B_{12}$ to grow, it can obtain the compound from bacteria and use it as a cofactor in a pathway for methionine biosynthesis that is thermal tolerant (*Kazamia et al., 2012*; *Xie et al., 2013*). Indeed, under elevated temperatures, $B_{12}$-providing bacteria increase the fitness of the alga (*Xie et al., 2013*). A mutualism was also observed between *C. reinhardtii* and *S. cerevisiae* in sealed microtiter plates, with the algae trading reduced nitrogen for $CO_2$ (*Hom and Murray, 2014*). While the significance of these interactions may be uncertain, they, and many yet to be discovered, likely shape the ways in which *C. reinhardtii* navigates in a complex biosphere.

## Conclusions

Although *C. reinhardtii* has been studied in the laboratory for many decades, we do not know the extent to which results from the laboratory reflect growth, life cycle and behavior of this alga in nature (*Box 2*). As a model system, *C. reinhardtii* is almost exclusively grown as a pure culture, a situation almost never encountered in the 'wild'. Returning a laboratory strain of *C. reinhardtii* to its native habitat would reveal whether domestication caused it to lose its ability to survive within the dynamic fabric of nature. Molecular analyses of the reintroduced strain could also reveal changes in the cells' physiology that underlie the loss of fitness in natural habitats, as well as other changes potentially associated with its adaptation to laboratory conditions.

Field surveys are often hampered by difficulties in assessing the metabolic state of the cells and in establishing key inter-organismal interactions. However, harnessing the full potential of meta-omics and single-cell technologies could provide a fuller appreciation of the physiological status of cells as they experience environmental fluctuations and the dominant interactions that shape the life of *C. reinhardtii*. Expanding this understanding will require time-resolved data on the geographical occurrence of *C. reinhardtii* in different habitats, its genetic potential and population genetics, and dissection of biotic and abiotic interactions. Such studies could then be extended to include analyses performed under controlled laboratory conditions that closely align with conditions encountered in the field, using innovative methods such as microfluidics to mimic conditions of the soil and other complex environments (*Stanley et al., 2016*).

### Acknowledgements

We thank Dr. Thomas Pröschold for helpful comments on this manuscript, Debbie Maizels for preparing *Figures 1* and *2*, Drs. Santosh Sathe and Pierre Durand for providing *Figure 3*, and Prasad Aiyar for providing *Video 1*. Because of space limitations, we were unable to cite many

worthy studies relevant to the topic of this article; we apologize to all of our colleagues whose work we did not discuss.

**Severin Sasso** is at the Matthias Schleiden Institute of Genetics, Bioinformatics and Molecular Botany, Friedrich Schiller University, Jena, Germany

severin.sasso@uni-jena.de

http://orcid.org/0000-0002-2226-845X

**Herwig Stibor** is in the Department Biology II, Ludwig Maximilian University, Munich, Germany

**Maria Mittag** is at the Matthias Schleiden Institute of Genetics, Bioinformatics and Molecular Botany, Friedrich Schiller University, Jena, Germany

**Arthur R Grossman** is at the Carnegie Institution for Science, Stanford, United States

arthurg@stanford.edu

*Author contributions:* Severin Sasso, Herwig Stibor, Maria Mittag, Arthur R Grossman, Drafting and revising the article

*Competing interests:* The authors declare that no competing interests exist.

**Funding**

| Funder | Grant reference number | Author |
|---|---|---|
| Deutsche Forschungsgemeinschaft | SFB 1127 | Severin Sasso Maria Mittag |
| U.S. Department of Energy | DE-FG02-07ER64427 | Arthur R Grossman |
| U.S. Department of Energy | DE-FG02-12ER16338 | Arthur R Grossman |
| Deutsche Forschungsgemeinschaft | SA 2453/1-1 | Severin Sasso |
| National Science Foundation | NSF-MCB 0951094 | Arthur R Grossman |

The funders had no role in study design, data collection and interpretation, or the decision to submit the work for publication.

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
