## [Decision Letter]

Thank you for submitting your article to *eLife* for consideration as a Feature Article. Your article has been reviewed by two peer reviewers, and the evaluation has been overseen by Stuart King as an Associate Features Editor and Peter Rodgers as the Senior Editor. The reviewers have opted to remain anonymous.

The reviewers have discussed the reviews with one another and the Associate Features Editor has drafted this decision to help you prepare a revised submission.

Summary:

This essay is being considered as part of a series of articles on "The Natural History of Model Organisms" (https://elifesciences.org/collections/8de90445/the-natural-history-of-model-organisms). Each article should explain how our knowledge of the natural history of a model organism has informed recent advances in biology, and how understanding its natural history can influence/advance future studies.

There is much that is interesting in this overview of the single-celled green alga *Chlamydomonas reinhardtii*. The reviewers recognize the difficulty of writing such a wide-ranging article and commend the authors on a thought-provoking piece. They believe that this article will serve the purpose of stimulating new questions in this area of research. However, a number of details should be attended to prior to publication.

Essential revisions:

1) The current title ("Hatched to be wild: *Chlamydomonas reinhardtii* in nature") does not represent the précis of the article. It is also inconsistent with the other articles in series. The editor will contact you separately to provide some more guidance in this area.

2) While the previous literature has often used "flagella", most working in this area have now migrated to the term "cilia". The "flagella" of *C. reinhardtii* are synonymous with the cilia or other eukaryotes and not at all with the flagella of bacteria. Given that this is an enormous contribution of *C. reinhardtii* research in recent decades, this should be properly developed in the opening paragraph. "Cilia" should then be used throughout the rest of the article. In the Abstract, please also add "biogenesis" or "assembly" (to "structure and function of cilia").

3) Figure 1 is a good addition to the article, yet the two panels are largely unrelated to each other and referenced in distant sections of the text. Please split this figure into two, such that each part can positioned closer to the relevant text for the benefit of the reader. In relation to point above, please also replace uses of "flagellum" with "cilium" etc.

4) While it is included as part of a list, referring to *C. reinhardtii* as a "preferred" model organism "for dissecting photosynthetic function […]" (Introduction) is somewhat of an understatement. Most of the key components were discovered in *Chlamydomonas* through the collection of acetate-requiring mutants. These components were subsequently re-discovered in *Arabidopsis*, often without proper attribution to the original decades-earlier discoveries. The history of the photosynthesis discoveries should not be lost by weak language. Later in the text, the contribution of the Hegemann lab to the field of optogenetics also seems understated with the term "leveraged" (Introduction).

5) In the paragraph that begins, "Studies of *C. reinhardtii* have enabled numerous landmark discoveries" (Introduction), the discoveries chosen are somewhat selective and, in the opinion of the reviewers, some may not be as "landmark" as others that are not cited. The reviewers would encourage the authors to consider which of their mentioned examples represent paradigm shifts or truly big leaps in the understanding of a process. In addition, the reviewers believe that following discoveries merit being considered for this section: the discovery of two related PSII reaction centre proteins (D1, D2); the placement of electron carriers in the Z-scheme by genetic analysis of specific mutants using absorption spectroscopy; and any of the dozens of papers on the centriole or the cell cycle (where *C. reinhardtii* made significant contributions to the field of cell biology).

6) The reviewers had several other comments related to citations. The authors acknowledge that: "Because of space limitations, we were unable to cite many worthy studies", but the following revisions could help to balance and complete the reference list. These revisions are mentioned below in the order they arise in the manuscript:

6a) There are too many citations for "CRISPR-mediated targeted gene disruptions" (Introduction) compared to all the other technologies in the preceding text.

6b) In the section on Genomics, it may be better to cite the revised chloroplast genome published by Gallaher et al., 2017, rather than the earlier 2002 genome. The Gallaher article also includes the transcriptome and updated gene models.

6c) There was a concern that the literature supporting the current section on chemotaxis is limited (subsection “Physiological and metabolic capabilities”). The general feeling was that findings based on, at best, a single published paper were not substantive contributions and did not warrant inclusion.

6d) No references are included to support the following statements: "the ability to grow under heterotrophic and fermentative conditions is clearly an adaptation to soil environments" (subsection “Physiological and metabolic capabilities”); "the number of competing algal species in the soil is likely to be fewer than that of lakes" (subsection “Biotic interactions”); "In the soil, ciliates such as Paramecium may represent a typical predator" (subsection “Biotic interactions”); that the palmelloid aggregates are a result of "failure" to hatch (subsection “Biotic interactions”). In each case, if suitable citations cannot be found, please consider softening the claims or providing more rationale for including these claims over alternative explanations (for example, couldn't heterotrophy and fermentation equally be adaptations to lake sediments? Why couldn't the palmelloid aggregates instead result from biofilm formation? Etc.)

7) It is worth noting that it is not just the "ability to grow in the dark" (Introduction), but rather to grow in the dark and retain a functional photosynthetic apparatus that made some aspects of the studies possible. Most plants will become yellow in the dark, because there is a photochemical requirement in the chlorophyll biosynthetic pathway. *Chlamydomonas* (and other algae as well as evergreens) have a second enzyme – protochlorophyllide oxidoreductase – that catalyses the same reaction without a requirement for a photon. This trait is important and should be mentioned in the text.

8) The fact that *C. reinhardtii* can "form gametes as part of the sexual reproduction cycle" is seemingly put forward as an advantage of this alga as a model organism (Introduction). The text should be revised to make it clearer that this trait is not unique, but rather universal among sexual organisms [which include almost all eukaryotes]). It is also not obvious what the authors mean by the phrase: "which allows for introduction of multiple traits (e.g. mutant alleles) in a single haploid strain". Does it refer to making stable diploids? Or the generating double and triple mutants through classical genetic crosses? Or something else? Please revise to make the meaning clearer.

9) There were some concerns that little is known about the snow algae and that the field remains more confused than this article would suggest (subsection “Habitats and biogeography of *C. reinhardtii”*). One reviewer noted that *Chlamydomonas nivalis* is perhaps better considered a cryotolerant mesophile, rather than a strict cryophile/psychrophile (see https://doi.org/10.2478/botcro-2013-0012 and https://doi.org/10.1111/1574-6941.12299), and that it is possibly not even a single species. Overall, the reviewers recommend that the authors acknowledge the confusion here and minimize the discussion, perhaps just to refer that some members of the genus are psychrophilic/cryotolerant and recognized occasionally by the patches of watermelon color on snow. [Notably, the related *Chloromonas nivalis* is more likely a true psychrophile but, because it belongs to a different genus, it is beyond the scope of this essay].

10) Some of the section on deflagellation is counter to the conclusion of the cited Quarmby, 2009 chapter (subsection “Physiological and metabolic capabilities”). The potential adaptions discussed in this essay are introduced in the book chapter, but the conclusion of the chapter is that they are less likely than deflagellation being a consequence of pathological over-stimulation of a disassembly pathway. The reviewers request that the authors re-read the chapter, and then revise this section (including its citations) accordingly.

11) It would be great if the authors could provide a video of their Figure 2B (like those in https://doi.org/10.1038/s41467-017-01547-8 or similar). These show the wolf pack feeding behavior and the deflagellation response very nicely.

12) There was much debate in the reviewer consultation about whether the word "pelagic" can refer to organisms living in freshwater lakes (in part because of its root from the Greek "pelagikos", which relates to the sea). Please revise the text to make it clearer whether sections are referring to oceanic/marine algae or lake-dwelling algae. If "pelagic" is still used, it should be clearly defined for the benefit of unfamiliar readers. Please consider something like "organisms that live in the water column of lakes and oceans, but not on or near the bottom of the lake or ocean".

13) In the text in Box 1. Taxonomy and laboratory history of *C. reinhardtii*, please add "from two parents or haplotypes" to the end of the sentence that reads "The sequencing of 39 common laboratory strains show that they fall into five genetically distinct lineages". This should be mentioned; otherwise it may sound like there are 5 haplotypes.

14) The concluding paragraph could be strengthened. For example, the sentence: "It would be interesting to examine the fitness and adaptation of a domesticated laboratory strain to its native environment." Why would this be interesting? Which strain? What questions would be addressed?

---

## [Author Response]

Essential revisions:1) The current title ("Hatched to be wild: Chlamydomonas reinhardtii in nature") does not represent the précis of the article. It is also inconsistent with the other articles in series. The editor will contact you separately to provide some more guidance in this area.

The title has been changed. It is now: "The Natural History of Model Organisms: From molecular manipulation of domesticated *Chlamydomonas reinhardtii* to survival in nature".

2) While the previous literature has often used "flagella", most working in this area have now migrated to the term "cilia". The "flagella" of C. reinhardtii are synonymous with the cilia or other eukaryotes and not at all with the flagella of bacteria. Given that this is an enormous contribution of C. reinhardtii research in recent decades, this should be properly developed in the opening paragraph. "Cilia" should then be used throughout the rest of the article. In the Abstract, please also add "biogenesis" or "assembly" (to "structure and function of cilia").

The term "flagella" has been replaced with "cilia" throughout the article, except for "intraflagellar transport", which appears to have been retained in the current literature. The biogenesis of cilia is now mentioned in the abstract. In the Introduction, the importance of ciliary research has been stressed; the relevant section is now: "Studies of *C. reinhardtii* have enabled numerous landmark discoveries and advances. One remarkable example is the discovery of intraflagellar transport of granule-like particles (Kozminski et al., 1993) and the roles of motors in the process (Prevo et al., 2017 and references therein). Furthermore, structural analyses of wild type and mutants defective for ciliary function/assembly (Goodenough and Heuser, 1985, Silflow and Lefebvre, 2001, Nicastro et al., 2006) have massively contributed to our knowledge of the building blocks of a cilium and their organization and function, and led to classic studies that demonstrated that aberrant ciliary function can cause human genetic diseases such as polycystic kidney disease (Pazour et al., 2000, Li et al., 2004)."

3) Figure 1 is a good addition to the article, yet the two panels are largely unrelated to each other and referenced in distant sections of the text. Please split this figure into two, such that each part can positioned closer to the relevant text for the benefit of the reader. In relation to point above, please also replace uses of "flagellum" with "cilium" etc.

This has been done.

4) While it is included as part of a list, referring to C. reinhardtii as a "preferred" model organism "for dissecting photosynthetic function […]" (Introduction) is somewhat of an understatement. Most of the key components were discovered in Chlamydomonas through the collection of acetate-requiring mutants. These components were subsequently re-discovered in Arabidopsis, often without proper attribution to the original decades-earlier discoveries. The history of the photosynthesis discoveries should not be lost by weak language. Later in the text, the contribution of the Hegemann lab to the field of optogenetics also seems understated with the term "leveraged" (Introduction).

As suggested, we have strengthened the language. The phrasing "preferred model" has been changed into "enormously powerful model" (Introduction). The sentence on optogenetics now reads: "The field of optogenetics experienced a quantum leap with the discovery of *C. reinhardtii* channelrhodopsins […]". Some other sentences in the Introduction have also been strengthened, too.

5) In the paragraph that begins, "Studies of C. reinhardtii have enabled numerous landmark discoveries" (Introduction), the discoveries chosen are somewhat selective and, in the opinion of the reviewers, some may not be as "landmark" as others that are not cited. The reviewers would encourage the authors to consider which of their mentioned examples represent paradigm shifts or truly big leaps in the understanding of a process. In addition, the reviewers believe that following discoveries merit being considered for this section: the discovery of two related PSII reaction centre proteins (D1, D2); the placement of electron carriers in the Z-scheme by genetic analysis of specific mutants using absorption spectroscopy; and any of the dozens of papers on the centriole or the cell cycle (where C. reinhardtii made significant contributions to the field of cell biology).

The following topics have been added to the section on landmark discoveries and current research areas (last paragraph of Introduction): the discovery of IFT (Kozminski et al., 1993), the elucidation of ciliary structure (Goodenough and Heuser, 1985, Silflow and Lefebvre, 2001, Nicastro et al., 2006), the placement of electron carriers in the Z-scheme (e.g. Gorman and Levine, 1965), the discovery of D1 and D2 (Chua and Gillham, 1977) and their inferred function as reaction centers (Deisenhofer et al., 1985; Trebst, 1986; Satoh et al., 2003), elucidation of the cell cycle (Cross and Umen, 2015) and the structure/function of basal bodies/centrioles (Dutcher and O'Toole, 2016).

6) The reviewers had several other comments related to citations. The authors acknowledge that: "Because of space limitations, we were unable to cite many worthy studies", but the following revisions could help to balance and complete the reference list. These revisions are mentioned below in the order they arise in the manuscript:6a) There are too many citations for "CRISPR-mediated targeted gene disruptions" (Introduction) compared to all the other technologies in the preceding text.

CRISPR-related references are now cited as follows: "(Ferenczi et al., 2017 and references therein)".

6b) In the section on Genomics, it may be better to cite the revised chloroplast genome published by Gallaher et al., 2017, rather than the earlier 2002 genome. The Gallaher article also includes the transcriptome and updated gene models.

We have kept the original references but added the recent work by Gallaher et al., 2018.

6c) There was a concern that the literature supporting the current section on chemotaxis is limited (subsection “Physiological and metabolic capabilities”). The general feeling was that findings based on, at best, a single published paper were not substantive contributions and did not warrant inclusion.

There are a number of papers on chemotaxis in *C. reinhardtii*. Chemotaxis to ammonium was described by Byrne et al., (1992); to nitrite by Ermilova and Zalutskaya, (2014); and to nitrate by Ermilova et al., (2009). To cover these three references more accurately, we have changed the citation to "Ermilova and Zalutskaya, 2014 and references therein".

6d) No references are included to support the following statements: "the ability to grow under heterotrophic and fermentative conditions is clearly an adaptation to soil environments" (subsection “Physiological and metabolic capabilities”); "the number of competing algal species in the soil is likely to be fewer than that of lakes" (subsection “Biotic interactions”); "In the soil, ciliates such as Paramecium may represent a typical predator" (subsection “Biotic interactions”); that the palmelloid aggregates are a result of "failure" to hatch (subsection “Biotic interactions”). In each case, if suitable citations cannot be found, please consider softening the claims or providing more rationale for including these claims over alternative explanations (for example, couldn't heterotrophy and fermentation equally be adaptations to lake sediments? Why couldn't the palmelloid aggregates instead result from biofilm formation? Etc.)

These issues have been addressed as follows:

The first sentence has been toned down. It now reads: "The ability of *C. reinhardtii* to grow under heterotrophic and fermentative conditions might be an adaptation to soil environments […]"

The sentence "the number of competing algal species in the soil […]" has been deleted.

The sentence on predators in the soil has been broadened. It now reads: "In the soil, animals such as earthworms or springtails, and protists are typical predators of microscopic algae (Schmidt et al., 2016, Seppey et al., 2017), but there is currently little specific information on predators of *C. reinhardtii*."

The sentence on palmelloids has been toned down, and a reference has been added: "These colonies are aggregates of *C. reinhardtii* that may form as a consequence of… (Khona et al., 2016)."

7) It is worth noting that it is not just the "ability to grow in the dark" (Introduction), but rather to grow in the dark and retain a functional photosynthetic apparatus that made some aspects of the studies possible. Most plants will become yellow in the dark, because there is a photochemical requirement in the chlorophyll biosynthetic pathway. Chlamydomonas (and other algae as well as evergreens) have a second enzyme – protochlorophyllide oxidoreductase – that catalyses the same reaction without a requirement for a photon. This trait is important and should be mentioned in the text.

This aspect has been rephrased as follows: "Another outstanding advantage of *C. reinhardtii* is its ability to grow in the dark on acetate-containing medium while retaining a functional photosynthetic apparatus, which enabled isolation of photosynthesis mutants even if they are light-sensitive (Levine, 1969, Spreitzer and Mets, 1980)." To keep the Introduction short and crisp, we have not included protochlorophyllide oxidoreductase, which seemed to us beyond the scope of this review.

8) The fact that C. reinhardtii can "form gametes as part of the sexual reproduction cycle" is seemingly put forward as an advantage of this alga as a model organism (Introduction). The text should be revised to make it clearer that this trait is not unique, but rather universal among sexual organisms [which include almost all eukaryotes]). It is also not obvious what the authors mean by the phrase: "which allows for introduction of multiple traits (e.g. mutant alleles) in a single haploid strain". Does it refer to making stable diploids? Or the generating double and triple mutants through classical genetic crosses? Or something else? Please revise to make the meaning clearer.

We have rephrased this sentence as follows: "Furthermore, sexual reproduction can be induced in the laboratory, which allows for the introduction of multiple traits in a single haploid strain (e.g. generating double or triple mutants)." The universality of sex in eukaryotes was not included at this point because the introductory section is quite long already, and we feel it is a general point that most people reading the manuscript will already know.

9) There were some concerns that little is known about the snow algae and that the field remains more confused than this article would suggest (subsection “Habitats and biogeography of C. reinhardtii”). One reviewer noted that Chlamydomonas nivalis is perhaps better considered a cryotolerant mesophile, rather than a strict cryophile/psychrophile (see https://doi.org/10.2478/botcro-2013-0012 and https://doi.org/10.1111/1574-6941.12299), and that it is possibly not even a single species. Overall, the reviewers recommend that the authors acknowledge the confusion here and minimize the discussion, perhaps just to refer that some members of the genus are psychrophilic/cryotolerant and recognized occasionally by the patches of watermelon color on snow. [Notably, the related Chloromonas nivalis is more likely a true psychrophile but, because it belongs to a different genus, it is beyond the scope of this essay].

This statement has been rephrased and is more general. It now reads: "[…] and some members of the genus *Chlamydomonas* are carotenoid-rich organisms present on the surface of snow, giving it a red appearance (Remias et al., 2005)."

10) Some of the section on deflagellation is counter to the conclusion of the cited Quarmby 2009 chapter (subsection “Physiological and metabolic capabilities”). The potential adaptions discussed in this essay are introduced in the book chapter, but the conclusion of the chapter is that they are less likely than deflagellation being a consequence of pathological over-stimulation of a disassembly pathway. The reviewers request that the authors re-read the chapter, and then revise this section (including its citations) accordingly.

This section has been revised. It now reads: "The biological reason for deciliation is still a mystery, but various hypotheses have been put forth (Quarmby, 2009). Deciliation is observed in a wide range of different cell types; for example, inhalation of irritant chemicals can lead to deciliation of respiratory epithelial cells in mammals (Buckley et al., 1984). Therefore, it seems likely that a predetermined break point is an ancient and inherent property of every cilium (Quarmby, 2009). Consequently, deciliation may not confer a selective advantage, but might be a consequence of pathological conditions that cause over-stimulation of the ciliary disassembly process. On the other hand, the ciliary membrane of *C. reinhardtii* is in direct contact with the environment (not protected by cell wall) and therefore, deciliation may reduce the entrance of noxious compounds into cells. Deciliation may also allow cells to escape when their cilia are stuck to the surface of a predator (Quarmby, 2009). Studying deciliation in the natural environment holds the promise of new insights into selection pressures that led to its evolution."

11) It would be great if the authors could provide a video of their Figure 2B (like those in https://doi.org/10.1038/s41467-017-01547-8 or similar). These show the wolf pack feeding behavior and the deflagellation response very nicely.

We have now reused the video previously published by Aiyar et al., (2017). This is now Video 1.

12) There was much debate in the reviewer consultation about whether the word "pelagic" can refer to organisms living in freshwater lakes (in part because of its root from the Greek "pelagikos", which relates to the sea). Please revise the text to make it clearer whether sections are referring to oceanic/marine algae or lake-dwelling algae. If "pelagic" is still used, it should be clearly defined for the benefit of unfamiliar readers. Please consider something like "organisms that live in the water column of lakes and oceans, but not on or near the bottom of the lake or ocean".

We have included an explanatory sentence as suggested: "The term 'pelagic zone' refers to the water column of lakes and oceans not on or near the lake or ocean bottom." In several cases, we have clarified that we are referring to the pelagic zone of lakes, not oceans.

13) In the text in Box 1. Taxonomy and laboratory history of C. reinhardtii, please add "from two parents or haplotypes" to the end of the sentence that reads "The sequencing of 39 common laboratory strains show that they fall into five genetically distinct lineages". This should be mentioned; otherwise it may sound like there are 5 haplotypes.

This has been added.

14) The concluding paragraph could be strengthened. For example, the sentence: "It would be interesting to examine the fitness and adaptation of a domesticated laboratory strain to its native environment." Why would this be interesting? Which strain? What questions would be addressed?

This sentence mentioned has been changed. It now reads: "Returning a *C. reinhardtii* laboratory strain to its native habitat would reveal whether domestication caused it to lose its ability to survive within the dynamic fabric of nature and allow for molecular analyses that could reveal changes in the cells' physiology responsible for the loss of their fitness in natural habitats, as well as other changes that may be associated with adaptation to laboratory conditions." Other parts of the concluding section have been revised as well.